# The Characteristic Function of Blood-Derived Exosomes and Exosomal circRNAs Isolated from Dairy Cattle during the Dry Period and Mid-Lactation

**DOI:** 10.3390/ijms241512166

**Published:** 2023-07-29

**Authors:** Yiru Shi, Zhengjiang Zhao, Xiao He, Junyi Luo, Ting Chen, Qianyun Xi, Yongliang Zhang, Jiajie Sun

**Affiliations:** Guangdong Provincial Key Laboratory of Animal Nutrition Control, College of Animal Science, South China Agricultural University, Guangzhou 510642, China; shiyiru00@163.com (Y.S.); zhengjiangzhao0314@163.com (Z.Z.); hexiao_0408@163.com (X.H.); luojunyi@scau.edu.cn (J.L.); allinchen@scau.edu.cn (T.C.); xiqianyun_scau@163.com (Q.X.)

**Keywords:** exosomes, mammary gland development, lactation, circular RNA, Sanger sequencing

## Abstract

Exosomes are key mediators of intercellular communication. They are secreted by most cells and contain a cargo of protein-coding genes, long noncoding RNAs (lncRNAs), and circular RNAs (circRNAs), which modulate recipient cell behavior. Herein, we collected blood samples from Holstein cows at days 30 (mid-lactation) and 250 (dry period) of pregnancy. Prolactin, follicle-stimulating hormone, luteinizing hormone, estrogen, and progesterone levels showed an obvious increase during D250. We then extracted exosomes from bovine blood samples and found that their sizes generally ranged from 100 to 200 nm. Further, Western blotting validated that they contained CD9, CD63, and TSG101, but not calnexin. Blood-derived exosomes significantly promoted the proliferation of mammary epithelial cells, particularly from D250. This change was accompanied by increased expression levels of proliferation marker proteins PCNA, cyclin D, and cyclin E, as detected by EdU assay, cell counting kit-8 assay, and flow cytometric cell cycle analysis. Moreover, we treated mammary epithelial cells with blood-derived exosomes that were isolated from the D30 and D250 periods. And RNA-seq of two groups of cells led to the identification of 839 differentially expressed genes that were significantly enriched in KEGG signaling pathways associated with apoptosis, cell cycle and proliferation. In bovine blood-derived exosomes, we found 12,747 protein-coding genes, 31,181 lncRNAs, 9374 transcripts of uncertain coding potential (TUCP) candidates, and 460 circRNAs, and 32 protein-coding genes, 806 lncRNAs, 515 TUCP candidates, and 45 circRNAs that were differentially expressed between the D30 and D250 groups. We selected six highly expressed and four differentially expressed circRNAs to verify their head-to-tail splicing using PCR and Sanger sequencing. To summarize, our findings improve our understanding of the key roles of blood-derived exosomes and the characterization of exosomal circRNAs in mammary gland development.

## 1. Introduction

Exosomes are nanoscale extracellular vesicles of endocytic origin [1] that are shed by most cell types and circulate in bodily fluids, such as blood [2], urine [3], saliva [4], and breast milk [5]. Exosomal contents comprise various growth factors, proteins, lipids, and nucleic acids [1]. In recent years, an increasing number of studies have attempted to understand the function of exosomes in mediating pathological and physiological processes [6]. Notably, studies on exosomes have been increasing remarkably in the recent years, especially following the discovery of functional miRNAs in exosomes [7]. The mammary gland, a dynamic organ that develops primarily in the adult, undergoes extensive expansion during puberty, followed by cycles of growth and regression with each estrus cycle and every pregnancy–lactation–involution cycle [8]. The complex regulation of mammary gland development has been extensively studied at the genetic [9], physiological [10], and morphological levels [8]. Exosomes evidently play a key role in different stages of mammary gland development; in particular, they are associated with the proliferation and differentiation of mammary gland cells [11].

Protein molecules or lipid ligands on exosomes directly activate receptors on the surface of target cells, generating signaling complexes and activating intracellular signaling pathways [12,13], or exosomes fuse with the plasma membrane of target cells or are endocytosed into cells, delivering their contents (i.e., proteins, lipids, and nucleic acids) and, consequently, regulating cell function and biological behavior [14]. Numerous circular RNAs (circRNAs) are enriched and stable in exosomes; they are detectable in blood and milk, and involved in intercellular information exchange [15]. In addition, circRNAs, as a novel class of endogenous noncoding RNAs, play a vital role in gene expression regulation, impacting many key physiological processes [16]. With the development of deep RNA sequencing (RNA-seq) technologies and novel bioinformatic approaches, recent studies have reported a large number of endogenous circRNAs that are abundant, stable, and ubiquitously expressed, with some circRNAs having unique roles [17,18]. The expression of circRNAs is closely related to the growth and development of the mammary gland in humans [19], rats [20], and sheep [21]. Furthermore, in dairy cattle, circRNAs are apparently involved in gene expression [22] and cell proliferation [23] in the mammary gland.

Herein, we determined the expression profile of exosomal circRNAs in bovine blood between the dry period (day 250 of pregnancy, D250) and mid-lactation stage (day 30 of pregnancy, D30) and attempted to identify circRNA candidates related to mammary gland development. Our findings should improve our understanding of the chief roles of blood-derived exosomes and circRNA candidates in mammary gland development and also provide a basis for the selection of circRNA candidates for more detailed functional studies for future research.

## 2. Results

### 2.1. Analysis of Blood-Derived Exosomes

We measured PRL, FSH, LH, E2, and PROG levels in blood of dairy cattle at D30 and D250. Relative to D30, blood FSH and PROG levels were highly significantly elevated at D250 (*p* < 0.01), while blood E2 level was significantly elevated (*p* < 0.05). In contrast, blood PRL and LH levels showed no significant differences between these groups (Figure 1A).

Blood-derived exosomes were identified and characterized by transmission electron microscopy, Zetasizer Nano analysis, and Western blotting. Consistent with previously reported results [24], transmission electron microscopy demonstrated that the size of the isolated particles ranged from 100 to 200 nm (Figure 1B). Western blotting revealed that these particles contained CD9 (motility-related protein-1), CD63 (LAMP-3, lysosome-associated membrane protein-3), and TSG101 (tumor susceptibility gene 101 protein) but not calnexin (Figure 1C), confirming successful isolation of exosomes from blood samples. On Zetasizer Nano analysis, a broad peak was observed, corresponding to the mean particle size of 160 nm (range, 40–400 nm) (Figure 1D).

### 2.2. Blood-Derived Exosomes Promote MAC-T Cell Proliferation

Western blotting showed that blood-derived exosomes promoted the expression of the proliferation marker proteins PCNA (proliferating cell nuclear antigen), cyclin D (cyclin protein D1), and cyclin E (cyclin protein E) (Figure 2A) in MAC-T cells. PCNA and cyclin E expression levels in the D30 and D250 groups were significantly higher than those in the negative control (NC) group. Further, cyclin D expression level in the D250 group was higher than that in the NC and D30 groups. EdU assay revealed that the fluorescence intensity of MAC-T cells treated with blood-derived exosomes was elevated; the intensity in the D250 group was higher than that in the D30 group (Figure 2B). CCK-8 assay was used to assess MAC-T cell proliferation at 12 h, 24 h, 36 h, and 48 h after treatment with PBS or blood-derived exosomes isolated from the D30 and D250 periods. The D250 and D30 groups showed significantly higher optical density values than the NC group, implying that the proliferation efficiency of MAC-T cells was significantly improved (Figure 2C). Flow cytometric cell cycle analysis showed that the proportion of cells in the S and G2/M phases was significantly higher in the D30 and D250 groups (Figure 2D). These results indicated that blood-derived exosomes isolated from the D30 and D250 periods promoted MAC-T cell proliferation, with those isolated from the D250 period having a more prominent effect.

### 2.3. RNA-Seq Analysis of MAC-T Cells Treated with Blood-Derived Exosomes

We constructed six cDNA libraries from MAC-T cells treated with blood-derived exosomes isolated from the D30 and D250 periods, and each library generated approximately 40.41 ± 3.06 million raw reads. After quality correction, each library comprised approximately 40.20 ± 3.07 million clean reads, accounting for 99.47% original reads (Appendix A). After removing all reads that mapped to a rRNA database, 39.95 ± 3.07 million remaining reads were subjected to assembly and gene abundance analyses (Appendix A). We compared all valid reads to the bovine reference genome and found that approximately 97.28% ± 0.21% valid reads could be successfully mapped to the genome (Appendix A). The distribution of total mapped reads in the reference genome was calculated; the exon region accounted for 86.21% of total mapped reads (Appendix A). These findings validated the successful construction of libraries and also their suitability for subsequent analyses. In total, 407 novel transcripts were identified across all six libraries (Appendix A), and transcript abundance was quantified using FPKM values (Appendix A). We also performed principal component analysis with FPKM values of the identified candidates and observed that differences between the D30 and D250 groups were much larger than those between experimental individuals (Figure 3A). In comparison with D250 libraries, we identified 839 significantly differentially expressed genes (Appendix A, Figure 3B), including 395 up- and 444 downregulated genes, in D30 libraries (Figure 3C). Differentially expressed transcripts were found to participate in 290 KEGG signaling pathways (Appendix A), including TNF signaling pathway, IL-17 signaling pathway, HIF-1 signaling pathway, and apoptosis (Figure 3D). KEGG pathway genes were subjected to gene set enrichment analysis (Appendix A); genes in the KO04668_TNF signaling pathway showed significant enrichment (Figure 3E).

### 2.4. Transcriptome Expression Analysis of Bovine Blood-Derived Exosomes

Overall, 96.32 ± 0.85 and 101.80 ± 1.97 million clean reads were obtained from the D30 and D250 groups, respectively (Appendix A). Clean reads were aligned to the bovine reference genome, and 63.87 ± 6.21 million (64.42% ± 5.19%) reads were successfully mapped to the genome (Appendix A). Approximately 2.05 ± 0.25 million clean reads that matched Ensembl protein-coding regions, accounting for 7.65% ± 0.96% clean reads in each library (Appendix A). In total, 53,302 assembled transcripts were identified across all libraries (Appendix A), including 31,003 novel lncRNAs and 9374 TUCP candidates. As evident from these results, annotation to the bovine reference genome was associated with poor outcomes, considering that a large number of novel transcripts needed more systematic annotation; besides, 31,181 lncRNA candidates, including 178 known lncRNAs, were identified after excluding annotation transcripts with protein-coding potentials (Figure 4A), and the expression levels of lncRNA candidates were much higher than those of the majority of mRNA molecules (Figure 4B). Further, we performed structural characterization by assessing exon size, fragment length, and open reading frame (ORF) length between lncRNAs and mRNAs (Figure 4C), as well as between mRNAs and TUCP candidates (Figure 4D). Consistent with the results of a previous study [25], the length of novel and annotated lncRNAs and their ORFs was shorter than that of mRNAs; moreover, lncRNAs contained fewer exons. However, the structural characteristics of TUCPs were similar to those of mRNAs.

### 2.5. Functional Enrichment Analysis of Differentially Expressed Transcripts in Bovine Blood-Derived Exosomes

We performed principal component analysis using FPKM values of all identified transcripts and found that differences caused by the physiological states of the mammary gland between the groups were much larger than those between experimental individuals (Figure 5A). With normalized FPKM, 32 differentially expressed Ensembl transcripts were found between D30 and D250 libraries (Appendix A), including 22 up- and 10 downregulated transcripts in the D250 group (Figure 5B). In addition, 806 differentially expressed lncRNA candidates were identified between the D30 and D250 groups (Appendix A), as well as 515 TUCP candidates (Appendix A). GO enrichment analysis indicated that differentially expressed Ensembl transcripts were significantly enriched in translation, ribosome assembly, and various snoRNA metabolic processes in the biological process category (Figure 5C; Appendix A). Furthermore, KEGG pathway enrichment analysis showed that these transcripts were significantly enriched in ribosome, regulation of actin cytoskeleton, RNA degradation, and RNA transport (Figure 5D; Appendix A).

### 2.6. Identification of circRNAs in Bovine Blood-Derived Exosomes

circRNA profiles were explored in bovine blood-derived exosomes. Overall, 460 circRNAs were identified in RNA-seq data using the find_circ and CIRI2 algorithms (Appendix A). Further, 256 circularization events were from exon regions, while 101 and 103 circularization events were from intergenic and intron regions, respectively (Appendix A). These events were found to originate from 324 hosting transcript loci, including 49 transcripts that generated multiple circRNA candidates. With normalized back-splice junction reads, we analyzed significant differences in circRNA candidates between D30 and D250 libraries (Appendix A). Forty-five circRNA candidates were differentially expressed (Figure 6A): 19 up- and 26 downregulated circRNAs were found in D250 libraries (Figure 6B). GO and KEGG pathway enrichment analyses were performed on hosting transcripts of the identified circRNA candidates (Appendix A). GO annotation indicated that hosting transcripts of significantly expressed circRNAs mainly participated in chromatin remodeling, peptidyl–serine phosphorylation, integrin-mediated signaling pathway, positive regulation of cytokinesis, and androgen receptor signaling pathway in the biological process category (Figure 6C). In addition, KEGG pathway enrichment analysis showed that these hosting transcripts participated in GnRH signaling pathway, ubiquitin mediated proteolysis, adherens junction, focal adhesion, regulation of actin cytoskeleton, Rap1 signaling pathway, and FoxO signaling pathway (Figure 6D).

### 2.7. RT-qPCR Analysis and Authentication of Blood-Derived Exosomal circRNAs

RT-qPCR was performed to validate the expression of 13 differentially expressed candidates between the D30 and D250 groups. As with our sequencing results, the expression level of all Ensembl transcripts, except RIPOR2 and FCGR3A, in the D250 group was higher than that of those in the D30 group. EIF3E and MRPS6 expression level in the D250 group was significantly upregulated than that in the D30 group (*p* < 0.05). Further, HDAC9 and RPL3 expression level in the D250 group was extremely higher than that in the D30 group (*p* < 0.01) (Figure 7A).

To validate the authenticity of the identified circRNAs, 10 circRNAs were subjected to RT-qPCR and Sanger sequencing. RT-qPCR successfully detected all circRNA candidates, and DNA sequencing confirmed the presence of head-to-tail splice junctions, as indicated by RNA-seq analyses (Figure 7B).

## 3. Discussion

The mammary gland is essential for lactation and responds to hormonal changes occurring during pregnancy and after birth [26]. Hormones, growth factors [27], and nutrient availability [28] closely control its development and functions [29]. During pregnancy, PROG, E2, cortisol, placental lactogen, and insulin must work in harmony to prepare this gland for lactation [30]. The initial growth of the lactiferous ductal system is dependent on E2, whereas growth hormone and cortisol have a synergistic effect [31]. Further, in the course of pregnancy, the increased levels of PRL, placental lactogen, E2, and PROG favor the development of the milk secretory alveolar apparatus [32]. In humans, serum PRL levels begin to linearly increase upon direct E2 stimulation during the first trimester of pregnancy [33]. Herein, we collected blood samples and measured serum concentrations of PRL, FSH, LH, E2, and PROG. As previously reported, hormone levels in the D250 group were generally higher than those in the D30 group. During a bovine lactation, the dry period is required to facilitate cell turnover in the bovine mammary gland to optimize milk yield in the next lactation [34]. It is notable that the aforementioned hormones can potentially inhibit mammary gland degeneration [35].

Mammary gland development is influenced by several noncoding RNAs and functional genes [36,37]. Exosomes are present in blood [38,39], and noncoding RNAs [40,41] and protein-coding genes [42] have been identified from exosomes. Exosomes play a key role in development [43], immunity [44], tissue homeostasis [45], cancer [46], and neurodegenerative diseases [47]. Nevertheless, the role of bovine blood-derived exosomes in regulating mammary gland development remains unclear. Exosome-encapsulated microRNAs reportedly function as circulating biomarkers for breast cancer [48], and serum-derived exosomal lncRNAs have therapeutic effects on breast cancer [49]. Moreover, exosomes are implicated in mammary gland development [50]. In this study, we collected blood samples from dairy cattle in D250 and D30, followed by isolation and identification of blood-derived exosomes. We treated MAC-T cells with these exosomes to explore their effects on the mammary gland. Our data indicated that blood-derived exosomes play an important role in mammary gland development by promoting the proliferation of MAC-T cells. Further, sequencing analyses of MAC-T cells treated with blood-derived exosomes from D250 and D30 revealed the participation of significantly differentially expressed genes in important biological processes, including cell proliferation, TNF signaling pathway [51], IL-17 signaling pathway [52], HIF-1 signaling pathway [53], apoptosis [54], FoxO signaling pathway [55], and p53 signaling pathway [56]. These functions may be regulated by cargo in blood-derived exosomes, such as functional proteins, metabolites, and nucleic acids, which play intercellular communication roles upon transfer to recipient cells [57].

Exosomes have a diverse composition of genetic material [58], including several circRNAs that are enriched and stable in them [15]. Therefore, we performed deep sequencing to comprehensively explore the profile of bovine blood-derived exosomal circRNAs, which led to the identification of 31,181 lncRNAs and 460 circRNAs. On comparing genomic features between lncRNAs and mRNAs, our results were consistent with those previously reported [59,60]. Cargo selection into exosomes is a regulated, non-random process; previous studies have indicated that RNAs are selectively encapsulated into exosomes [61]. In this study, we found that relative to mRNAs, lncRNAs encapsulated in bovine blood-derived exosomes were expressed at higher levels, which suggests that they play a comparatively crucial role in physiological function [62]. The circular structure of circRNAs makes them more stable than other RNA types [16]. Many circRNAs are conserved in mammals and have potential biological functions [63]. Of the 460 identified circRNAs, 45 were differentially expressed between the D250 and D30 groups. circRNAs were recently suggested to play a key role in dairy cattle immunity, development, and disease [64]. Herein, functional enrichment analyses showed that hosting transcripts of the identified circRNA candidates participated in cell cycle and proliferation, Rap1 signaling pathway, focal adhesion, regulation of actin cytoskeleton, and FoxO signaling pathway. Based on our findings, we suggest that these predicted circRNAs enveloped in blood-derived exosomes are an attractive regulatory element in cell proliferation and mammary gland development.

To the best of our knowledge, this is the first study to identify circRNAs in bovine blood-derived exosomes, although their presence in human serum exosomes [65], porcine milk exosomes [24], and bovine milk exosomes [66] has been reported previously. Our results present comprehensive information on the expression profiles of circRNAs in bovine blood-derived exosomes and should considerably enrich the transcript genomic database of dairy cattle.

To conclude, we successfully isolated and verified blood-derived exosomes from dairy cattle, and these exosomes were found to promote MAC-T cell proliferation, particularly those from D250. We compared exosomal RNA expression in bovine blood between the D250 and D30 groups and found numerous lncRNAs and circRNAs in blood-derived exosomes. Functional enrichment analyses revealed that hosting transcripts of the identified circRNA candidates were significantly enriched in various key processes, including apoptosis, cell cycle, and proliferation. Our findings improve our understanding of the important roles of blood-derived exosomes and the characterization of circRNA candidates in mammary gland development.

## 4. Materials and Methods

### 4.1. Animals and Management

We selected six healthy mastitis-free Holstein cows from the Wenshi Dinghu dairy farm (Zhaoqing, China). Three cows were in D30, while the other three were in D250. Approximately 200 mL blood samples from each cow were collected via the tail vein, and serum concentrations of prolactin (PRL), follicle-stimulating hormone (FSH), luteinizing hormone (LH), estrogen (E2), and progesterone (PROG) were measured using ELISA kits (Jiancheng Bioengineering Institute, Nanjing, Jiangsu, China). All experiments were performed in accordance with the procedures approved by the Institutional Animal Care and Use Committee of the South China Agricultural University (ethics approval code: SCAU2018F006, 13 March 2018).

### 4.2. Exosome Isolation

Milk exosomes were separated, as previously described [25]. Briefly, blood samples obtained from dairy cattle were centrifuged at 3000× *g* and 4 °C for 15 min; the supernatant thus obtained was filtered through a 0.22-μm filter (Millipore, Billicera, MA, USA) to obtain a clear fraction. This fraction was then ultracentrifuged at 120,000× *g* and 4 °C for 3 h using an SW70Ti rotor (Beckman Coulter Instruments, Fullerton, CA, USA), and the pellets were re-suspended in PBS to obtain exosome solutions. These solutions were stored at −80 °C for further analyses.

### 4.3. Transmission Electron Microscopy and Particle Size Analysis

Approximately 10 mL purified exosomal fractions were analyzed under a transmission electron microscope (Talos F200S 1604154S, Hillsboro, OR, USA). Samples were placed on formvar-coated copper grids for 2 min, briefly washed with ultrapure water, negatively stained with 1% uranyl acetate, and then observed by transmission electron microscopy. Size distribution was measured using Zetasizer Nano ZS-90 (Malvern, Great Malvern, UK).

### 4.4. Western Blotting Analysis of Exosomal Biomarkers

To confirm the presence of bovine blood-derived exosomes, three exosomal biomarkers (tumor susceptibility gene 101 protein (TSG101), CD9, and CD63) were subjected to Western blotting. Exosomal protein content was assayed using the Pierce^®^ BCA Protein Assay Kit (Jiancheng Institute of Bioengineering, Nanjing, China). Proteins (20 μg) were first separated by 10% sodium dodecyl sulfate–polyacrylamide gel electrophoresis and then transferred to a polyvinylidene difluoride membrane (Millipore, Bedford, MA, USA). After blocking with 5% skimmed milk for 2 h, the membranes were incubated overnight at 4 °C with calnexin, TSG101, CD9, and CD63 antibodies (ZenBio, Chengdu, China). Subsequently, they were incubated with the secondary antibody horseradish peroxidase-conjugated goat anti-rabbit IgG (Sangon Biotech, Shanghai, China) for 1 h at 25 °C. Finally, enhanced chemiluminescence luminous fluid (Solarbio Life Sciences, Beijing, China) was used for band visualization; protein bands were quantified by densitometry using ImageJ v1.8 (National Institutes of Health, Bethesda, MD, USA).

### 4.5. MAC-T Cell Culture

The bovine mammary epithelial cell line MAC-T was purchased from the Chinese Collection of Authenticated Cell Cultures (Beijing, China). Cells were cultured in Dulbecco’s modified Eagle’s medium (Gibco, Grand Island, NY, USA) containing 10% fetal bovine serum (Gibco) and 1% penicillin–streptomycin (Invitrogen, Carlsbad, CA, USA) in a humidified incubator at 37 °C and 5% CO_2_.

### 4.6. Western Blotting Analysis of MAC-T Cells

MAC-T cells were lysed, and total proteins were extracted using radioimmunoprecipitation assay reagent (Thermo Scientific, Waltham, MA, USA) containing protease inhibitors. All extracted proteins were diluted with sodium dodecyl sulfate buffer and boiled at 95 °C for 10 min. Equal amounts of proteins were separated by sodium dodecyl sulfate–polyacrylamide gel electrophoresis and transferred to a polyvinylidene fluoride membrane (Millipore, Bedford, MA, USA). After blocking with 5% skimmed milk for 2 h at room temperature, the membranes were probed with primary antibodies to detect cyclin D, cyclin E, and proliferating cell nuclear antigen (PCNA) (ZenBio), followed by incubation with goat anti-rabbit HRP conjugate antibody for 30 min at 37 °C (Sangon Biotech). Finally, enhanced chemiluminescence luminous fluid (Solarbio Life Sciences) was used for band visualization; protein bands were quantified by densitometry using ImageJ v1.8 (National Institutes of Health).

### 4.7. EdU Assay

MAC-T cells were seeded in 6-well culture plates at a density of 1 × 10^5^ cells/well and treated with PBS or blood-derived exosomes isolated from the D30 and D250 periods (50 μg/mL) for 24 h. Subsequently, EdU staining was performed, according to the manufacturer instructions (Beyotime Biotechnology, Shanghai, China).

### 4.8. Cell Counting Kit-8 (CCK-8) Assay

MAC-T cells were seeded in a 96-well culture plate (1 × 10^3^ cells per well) and treated with PBS or blood-derived exosomes isolated from the D30 and D250 periods (50 μg/mL). The end of 3 h of treatment was recorded as 0 h. CCK-8 reagent (EZBioscience, Roseville, MN) was added at 0 h, 12 h, 24 h, 36 h, and 48 h, followed by incubation for approximately 1 h. Optical density of each well was then measured at 450 nm using a multifunctional microplate reader (Thermo Fisher Scientific, Waltham, MA, USA).

### 4.9. Flow Cytometric Cell Cycle Analysis

MAC-T cells were treated with PBS or blood-derived exosomes isolated from the D30 and D250 periods (50 μg/mL). After 48 h, cells were collected, fixed in 75% ethanol, and stored overnight at −20 °C. They were then resuspended in 500 μL PI/RNase staining buffer (BD Biosciences, Franklin Lakes, NJ, USA) and incubated at 37 °C for 30 min. Flow cytometric cell cycle analysis was performed on the BD Accuri™ C6 flow cytometer and FACSDiVa software v9.0 (BD Biosciences).

### 4.10. RNA Extraction and Sequencing of MAC-T Cells Treated with Blood-Derived Exosomes

Cells were seeded in a 6-well culture plate (1 × 10^5^ cells/well) and treated with blood-derived exosomes at D30 and D250 (50 μg/mL) for 48 h. TRIzol (Invitrogen) was used for total RNA extraction from MAC-T cells, according to the manufacturer’s instructions. Subsequently, eukaryotic mRNA was enriched by oligo-dT beads. This enriched mRNA was then fragmented into short segments using a fragmentation buffer and reverse transcribed into cDNA using the NEBNext^®^ Ultra RNA Library Prep Kit (New England Biolabs, Ipswich, MA, USA). The resulting cDNA library was sequenced using Illumina Novaseq 6000 by Gene Denovo Biotechnology Co., Ltd. (Guangzhou, China).

Raw reads obtained on sequencing were further filtered by fastq (v0.18.0) [67] to obtain high-quality clean reads; this process involved removing reads containing adapters, reads containing > 10% unknown nucleotides, and low-quality reads containing > 50% low-quality (Q value ≤ 20) bases. Bowtie2 v2.2.8 [68] was used for mapping reads to a rRNA database, and unmapped reads were subjected to further analyses. An index of bovine reference genome (*bosTau* 9) was built, and the remaining clean reads were aligned to this reference genome using Hisat2 v2.0.4 [69]. The mapped reads were assembled by StringTie v1.3.1 and annotated with Ensembl transcript regions in a reference-based approach [70]. For each transcription region, a fragment per kilobase of transcript per million mapped reads (FPKM) value was calculated to quantify its expression abundance using RSEM [71], and differential expression analysis was performed by DESeq2 [72]. Transcripts with false discovery rate < 0.05 and absolute fold change ≥ 2 were considered to be differentially expressed. Further, Kyoto Encyclopedia of Genes and Genomes (KEGG) pathway enrichment analysis was performed to identify significantly enriched metabolic or signal transduction pathways for differentially expressed genes and to compare differentially expressed genes with the whole genome background. We also performed gene set enrichment analysis to identify whether a set of genes in specific KEGG pathways showed significant differences in two groups [73].

### 4.11. Blood-Derived Exosomal RNA Preparation and Sequencing

We used TRIzol (Invitrogen) to extract total RNA from blood-derived exosomes, as per the manufacturer’s instructions. To remove genomic DNA, RNA samples were treated with DNase I (Takara, Dalian, China). RNA samples with RNA integrity number ≥ 7.5 were further analyzed. In each experimental group, two randomly selected samples were mixed in equal quantities to derive one sequenced pool. The Ribo-Zero^TM^ rRNA Removal Kit (Illumina, San Diego, CA, USA) was used to remove rRNA from RNA libraries, and the remaining RNA fragments were reverse transcribed to cDNA using the RNA-seq Library Preparation Kit (Illumina). Finally, paired-end sequencing of the libraries was performed on an Illumina HiSeq™ 4000 sequencer (LC Bio, Hangzhou, China).

### 4.12. Transcriptome Analysis of Blood-Derived Exosomes

Raw reads were processed using in-house Perl scripts. Clean reads were obtained by eliminating reads containing adapters, reads containing > 10% unknown nucleotides, and low-quality reads containing > 50% low-quality (Q value ≤ 20) bases. Further, Q20, Q30, and GC contents were calculated. The index of the reference genome was built using Bowtie2 v2.2.8, and paired-end clean reads were aligned to the reference genome (*bosTau* 9) using Hisat2 v2.0.4 [68]. The mapped reads were assembled by StringTie v1.3.1 and annotated with Ensembl transcript regions in a reference-based approach [70]. Transcript expression levels were measured and normalized as FPKM. Ballgown v2.20.0 [74] was used to compare differentially expressed transcripts and generate tables and plots.

### 4.13. Identification of Unannotated Transcripts and circRNA Candidates

To identify novel long noncoding RNAs (lncRNAs) and transcripts of uncertain coding potential (TUCP), we followed these steps: (1) unannotated transcripts with one exon were deleted; (2) unannotated transcripts with a length of <200 bp were removed; (3) unannotated transcripts that overlapped with the exon region of database-annotated RNAs were removed using Cuffcompare [75] (meanwhile, transcripts overlapped with known lncRNA exon regions in the database as annotated lncRNAs); (4) unannotated transcripts with FPKM < 0.5 were eliminated; and (5) unannotated transcripts that did not pass the protein-coding-score test were removed using CPC [76], CNCI [77], and Pfam-scan [78]. Transcripts that passed the size and abundance filters of our pipeline and with a Pfam domain hit were defined as TUCP set transcripts [79].

To identify circRNAs, RNA-seq data were analyzed using the find_circ [80] and CIRI2 [81] algorithms. circRNA candidates identified by both these algorithms were subjected to further evaluation. The expression levels of circRNA candidates were calculated with back-splice junction reads, followed by normalization with the TMM algorithm. The edgeR package v3.30.3 [82] was then used to identify differentially expressed circRNAs with false discovery rate < 0.05.

### 4.14. Functional Enrichment Analysis of Differentially Expressed Transcripts and circRNA-Hosting Genes

Differentially expressed transcripts and circRNA-hosting genes were subjected to gene ontology (GO) enrichment analysis with the Goseq R package [83], and KEGG pathway enrichment analysis was conducted using KOBAS [84].

### 4.15. RT-qPCR

Total RNA was extracted from bovine blood-derived exosomes by TRIzol (Invitrogen), and 500 ng total RNA was reverse transcribed using a kit (Vazyme, Nanjing, China) to obtain cDNA. Primer Premier 5.0 was used to design quantitative primers for differentially expressed transcripts between the D30 and D250 groups. Primer sequences are listed in Appendix A. The external reference was cel-miR-39-3p (RiboBio, Guangzhou, China). The reaction mixture comprised 1 μL cDNA, 0.5 μL of upstream and downstream primers, 10 μL SYBR Green Real-Time PCR Master Mix (Vazyme, Nanjing, China), and 8 μL ddH_2_O. The cycling conditions were as follows: pre-denaturation at 95 °C for 10 min, followed by 40 cycles at 95 °C for 15 s, 60 °C for 15 s, and 72 °C for 40 s. After amplification, the dissociation curve was analyzed. Relative quantification was performed using the 2^−ΔΔCT^ method.

### 4.16. Validation of circRNAs by Sanger Sequencing

To verify the circular structure of circRNAs, we designed a pair of divergent primers and verified their head-to-tail splicing using PCR and Sanger sequencing. Primer sequences are listed in Appendix A. PCR was performed on a Bio-Rad system (Bio-RAD, Hercules, CA, USA) in a 20 μL reaction volume containing 2 μL cDNA, 7 μL nuclease-free water, 1 mM of each primer, and 10 μL of 2× Tap Master Mix (Vazyme). The cycling conditions were as follows: 95 °C for 3 min, followed by 30 cycles at 95 °C for 30 s, 30 s at the corresponding annealing temperature, and 72 °C for 30 s, and final elongation at 72 °C for 10 min. PCR products of 10 randomly selected circRNAs (six highly expressed and four differentially expressed) were confirmed by agarose gel electrophoresis.

### 4.17. Statistical Analysis

Values represent mean ± SEM. Data were analyzed using SPSS 17.0 (SPSS Inc., Chicago, IL, USA). One-way ANOVA was used to analyze the function of exosomes in MAC-T cells subjected to EdU assay, CCK-8 assay, and flow cytometric cell cycle analysis. We used *t*-tests to compare hormone levels in blood samples collected from the D30 and D250 groups, as well as relative gene expression in exosomes. *p* < 0.05 indicated statistically significant values, and *p* < 0.01 indicated extremely significant values. GraphPad Prism v8.0 was used to construct charts.

## Figures and Tables

**Figure 1 ijms-24-12166-f001:**
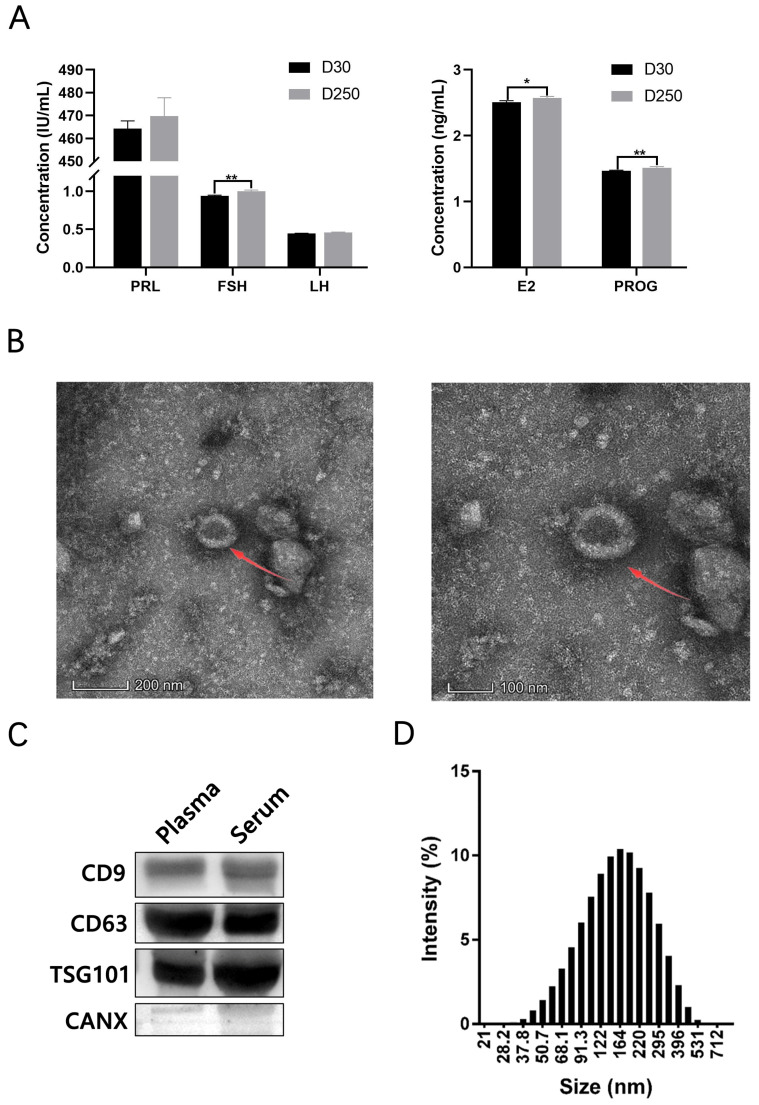
Identification of exosomes in bovine blood. (**A**) Hormone levels measured in blood from dairy cattle on D30 and D250. (**B**) Transmission electron microscopy images of blood-derived exosomes. Arrows point toward bovine blood-derived exosomes. (**C**) Exosome-specific markers CD9, CD63, and TSG101 and the non-exosomal marker calnexin detected by Western blotting. (**D**) Size distribution of blood-derived exosomes measured by Zetasizer Nano analysis. The asterisks above the graph bar indicate statistically significant differences according to Student’s *t*-test (* *p* < 0.05, ** *p* < 0.01). Mean relative expression levels, standard error and *p*-values were calculated using the software SPSS 17.0.

**Figure 2 ijms-24-12166-f002:**
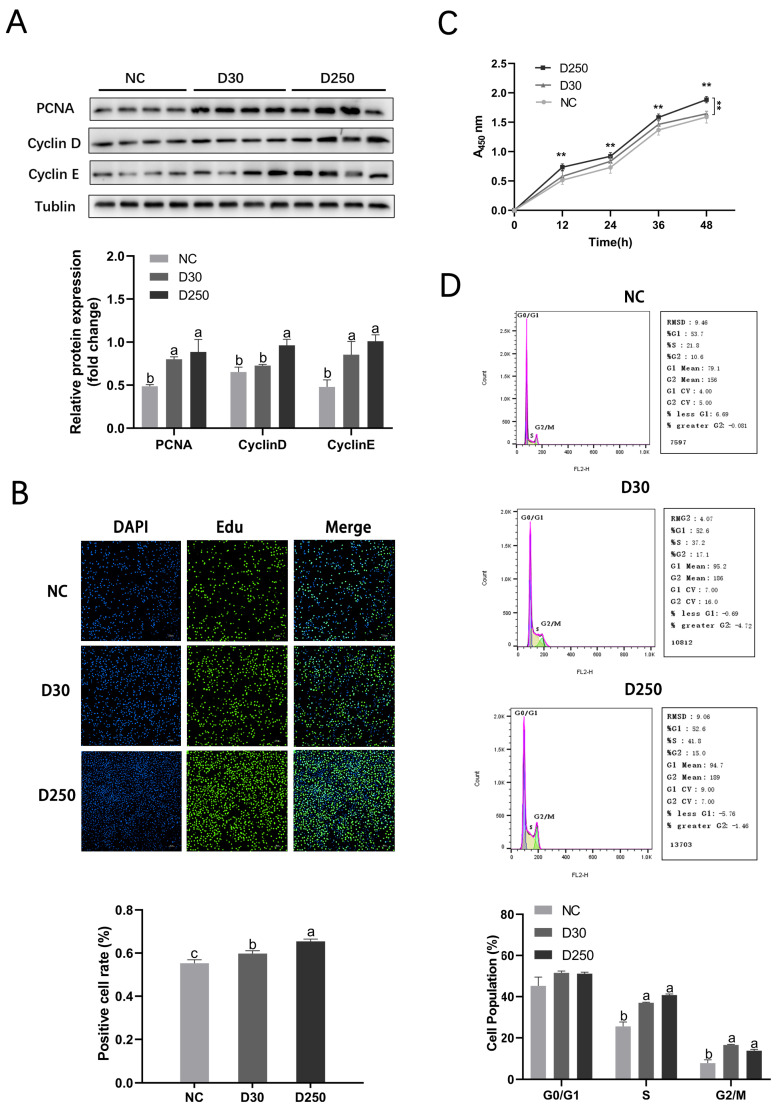
Proliferation of MAC-T cells induced by blood-derived exosomes in vitro. (**A**) Western blotting showing the expression levels of cell proliferation-related proteins in the NC, D30, and D250 groups. (**B**) EdU assay with MAC-T cells. (**C**) CCK-8 assay with MAC-T cells treated with PBS or blood-derived exosomes isolated from the D30 and D250 periods. (**D**) Flow cytometric cell cycle analysis. The asterisks above the graph bar indicate statistically significant differences according to Student’s *t*-test (** *p* < 0.01). The different lowercase letters above the graph bar indicate statistically significant differences between two groups. Mean relative expression levels, standard error and *p*-values were calculated using the software SPSS 17.0.

**Figure 3 ijms-24-12166-f003:**
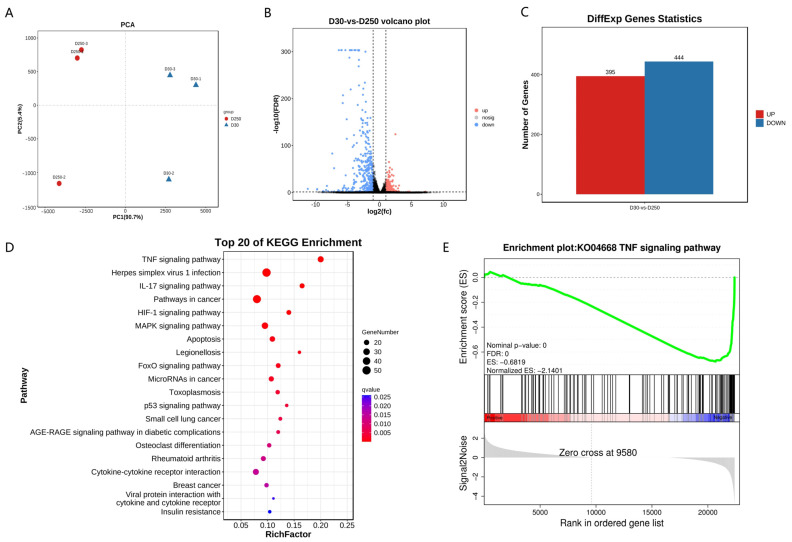
Functional enrichment analysis. (**A**) Principal component analysis based on whole transcripts in MAC-T cells treated with blood-derived exosomes isolated from the D30 and D250 periods. (**B**) Volcano plot showing significantly differentially expressed genes. (**C**) Statistics pertaining to differentially expressed genes. (**D**) KEGG pathway enrichment analysis of differentially expressed genes. (**E**) Gene set enrichment analysis plot for the KO04668_TNF signaling pathway.

**Figure 4 ijms-24-12166-f004:**
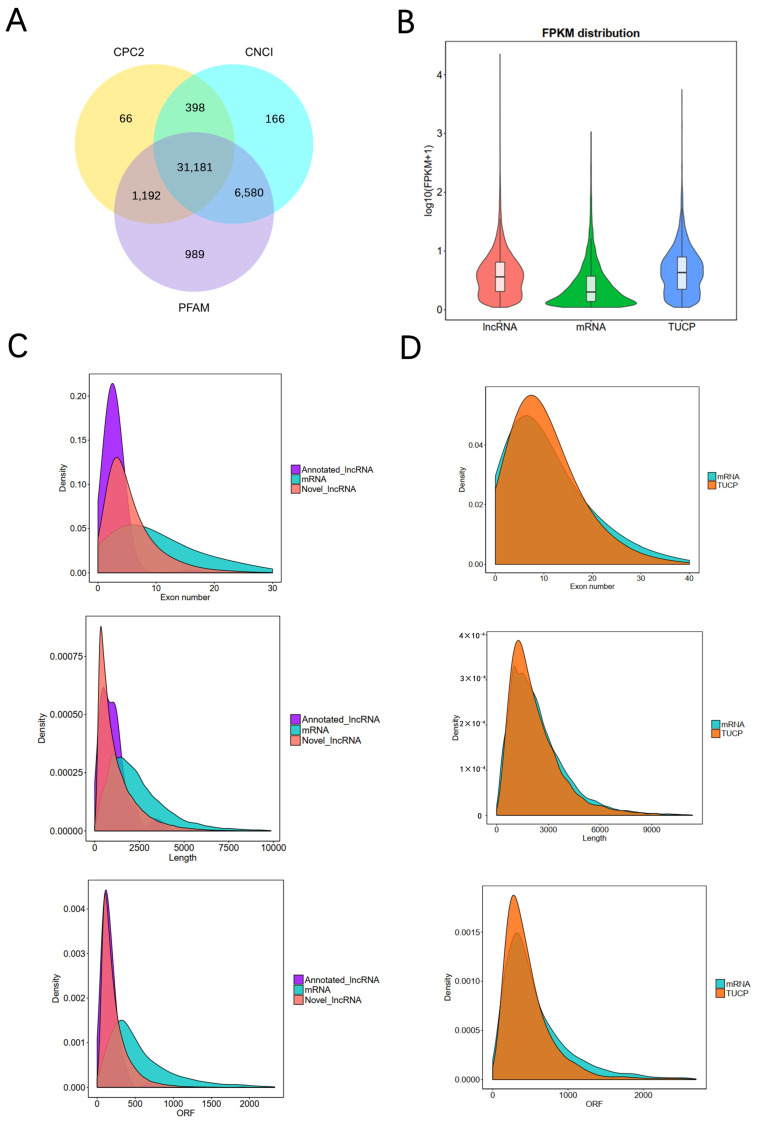
Identification and characterization of lncRNAs in bovine blood-derived exosomes. (**A**) Selection of lncRNAs in bovine blood-derived exosomes using CPC, Pfam-scan, and CNCI; 31,181 novel lncRNAs were identified after eliminating putative protein-coding transcripts. (**B**) Expression level of mRNA, TUCP, and lncRNA based on log_10_(FPKM + 1). (**C**) Distribution of exon density in lncRNAs and mRNAs, distribution of lncRNA and mRNA length, and distribution of ORF density in lncRNAs and mRNAs. (**D**) Distribution of exon density in TUCP and mRNAs, distribution of TUCP and mRNAs length, and distribution of ORF density in TUCP and mRNAs.

**Figure 5 ijms-24-12166-f005:**
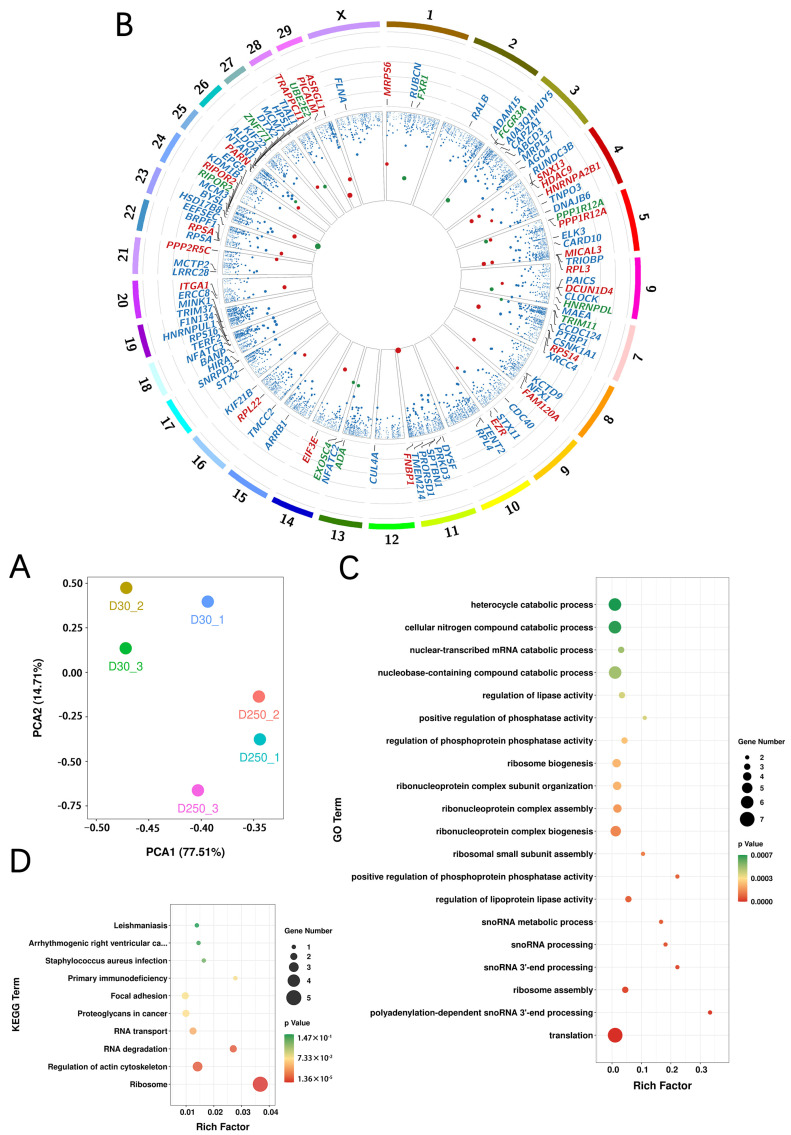
Cluster and enrichment analyses of exosomal transcripts. (**A**) Principal component analysis of whole transcripts in blood-derived exosomes isolated from the D30 and D250 periods. (**B**) Circos plot showing gene distribution on the chromosomes. (**C**) GO annotation analysis. (**D**) KEGG pathway enrichment analysis of neighboring gene functions.

**Figure 6 ijms-24-12166-f006:**
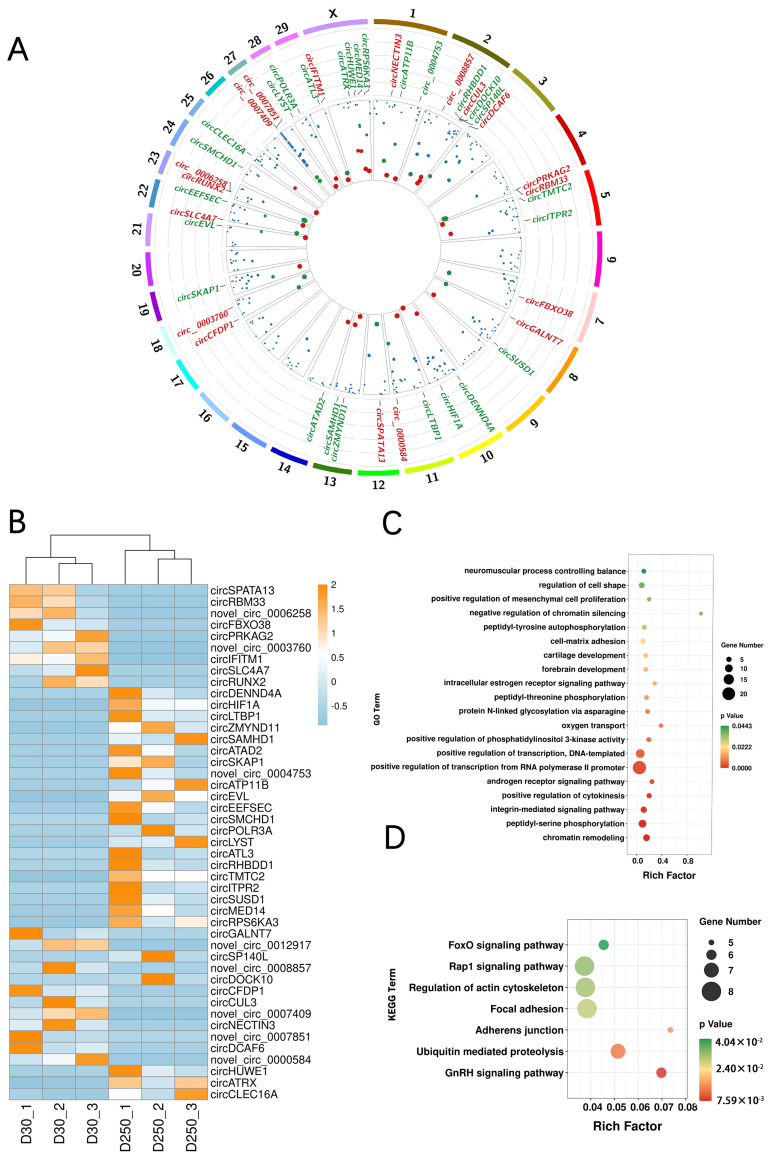
Cluster and enrichment analyses of bovine blood-derived exosomal circRNAs. (**A**) Circos plot showing the distribution of bovine blood-derived exosomal circRNAs on the chromosomes. (**B**) Heatmap of differentially expressed circRNAs between the D30 and D250 groups. (**C**) GO and (**D**) KEGG pathway enrichment analyses.

**Figure 7 ijms-24-12166-f007:**
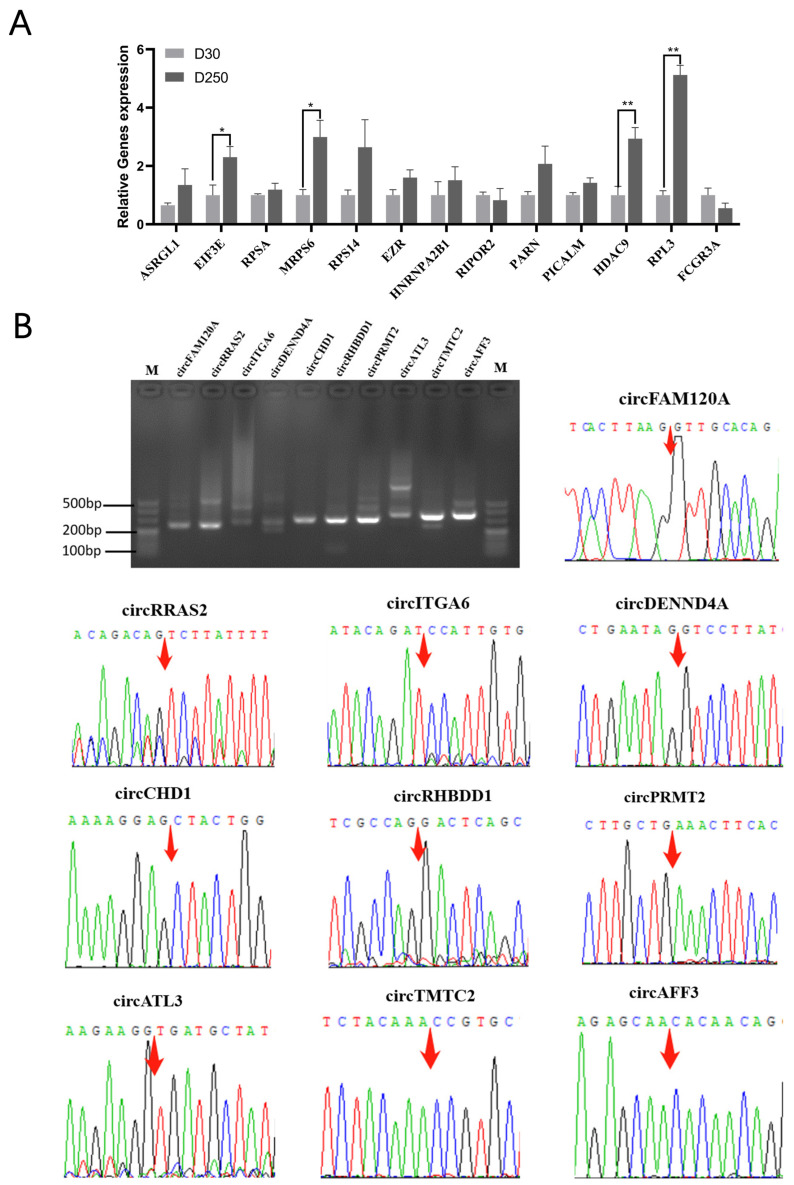
Validation of the presence of few selected blood exosomal circRNAs. (**A**) Relative expression levels of genes in blood-derived exosomes isolated from the D30 and D250 periods. (**B**) RT-qPCR amplimers derived from circRNAs using divergent primers for bovine blood-derived exosome RNA; head-to-tail splice junctions for circRNAs were confirmed by DNA sequencing and are marked with a red arrow on the chromatograms. The asterisks above the graph bar indicate statistically significant differences according to Student’s *t*-test (* *p* < 0.05, ** *p* < 0.01). Mean relative expression levels, standard error and *p*-values were calculated using the software SPSS 17.0.

## Data Availability

Not applicable.

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
