# Peer review of "The Characteristic Function of Blood-Derived Exosomes and Exosomal circRNAs Isolated from Dairy Cattle during the Dry Period and Mid-Lactation"

_ijms, 2023, doi:10.3390/ijms241512166_

Round 1

Reviewer 1 Report

This paper details a comprehensive study into the role of exosomes during dairy cow lactation and pregnancy.  This study was very interesting and very well presented.  I enjoyed reading this publication and as such have minor comments and recommendations.

While the layout of this journal places the results ahead of the materials and methods, the full definitions of the abbreviated proteins in the beginning of the results section would make this an easier read.

With regard to research design, it would have been nice to have performed the experiment on the same cows at day 30 and day 250 but I appreciate the logistics would have been difficult.

In the methods and materials there is no indication of how much blood was collected from each trial cow.

Line 466:  the full name and location for the company 'Vazyme' as per line 461 is needed.

Line 466: the '2' in H2O needs to be subscript.

Overall the figures are a bit rough, higher resolution figures would be nice but not necessary as they do provide all the information necessary.

Author Response

Dear reviewer,

Thank you for offering us an opportunity to improve the quality of our submitted manuscript (ijms-2497396). We appreciated very much the reviewers’ constructive and insightful comments. In this revision, we have addressed all of these suggestions. We hope the revised manuscript has now met the publication standard of your journal.

We highlighted all the revisions in red colour.

Our point-to-point responses to the queries raised by the reviewers are listed in the attachment.

Reviewer 2 Report

The authors investigated the expression profile of bovine blood-derived exosomes at two different stages of the reproductive cycle. They mainly focused on the identification of differentially expressed circRNAs in these exosomes. In a first step, they assessed the biological activity of their exosome preparation on MAC-T cells and sequenced the transcriptomes at days 30 and 250. The results support a measureable and differentiable effect of the exosomes isolated at two time points on the proliferation of the cells. In a next step, the transcriptomes of the isolated exosomes were analyzed. Among the identified exosome transcripts, 460 circRNAs were predicted with 45 differentially expressed at day 30 vs. day 250. Finally, the differential expression of 13 transcripts giving the basis for circRNAs was investigated with RT-qPCR, and 10 circRNAs were sequenced to confirm the circularization events.

In general, this is a very comprehensive and straightforward study. It indicates a potential role for exosomal circRNAs in cross-tissue communication.

However, some points should be considered in a revision:

(I) A biological role of exosomal components was shown only indirectly by exposing MAC-T cells to exosomes isolated at different time points. Therefore, the title of the manuscript is somewhat misleading indicating an investigation on the role of circRNAs at different lactation stages, which was not performed. The title and the statements in the abstract and the discussion should consider this fact.

(II) Table S2E should contain “Novel genes identified in all libraries” from the MAC-T cells. However, there are several genes listed with reference to Bos taurus. They are apparently not novel.

(III) Table S2F contains only “D250-3_fpkm” but D250-1/2_fpkm is missing. Maybe due to this fact, the table contains numerous genes with fpkm = 0 and count = 0 in all columns. These genes couldn’t be annotated in this study.

(IV) Most surprisingly is the extreme high number of transcripts in the blood exosomes. Is it realistic that with > 12,000 coding transcripts more than half of the bovine genes is shed into circulation via exosomes? Isn’t it likely that there is a substantial role of exosomes in transport of components from dead cells? Moreover, the enrichment of mitochondrial genes, components of the ribosomal machinery, actins and collagens does not reflect physiological processes. The authors should discuss this phenomenon.

(V) There is currently a hype about the role of exosomal micro RNAs. The authors did not even mention this fact in their introduction.

There are some minor spelling errors and argumentative issues (e.g. l24-26: make two sentences out of it; l252-253 “D250 is required to facilitate cell turnover…???), which should be edited.  

Author Response

(The authors gave the same response as above.)

Round 2

Reviewer 2 Report

The authors have addressed all of my comments in an appropriate manner. This makes the manuscript acceptable now.